# Prevalence and factors which influence early antenatal booking among women of reproductive age in Tanzania: An analysis of data from the 2015-16 Tanzania Demographic Health Survey and Malaria Indicators Survey

**Fabiola V. Moshi** *

Department of Nursing Management and Education, School of Nursing and Public Health, The University of Dodoma, Dodoma, Tanzania

* fabiola.moshi@gmail.com

## Abstract

### Background

Early initiation of the antenatal clinic is vital as it allows early detection, management, and prevention of problems that may occur during pregnancy time. The analysis aimed to determine the prevalence and factors which influence early antenatal booking among women of reproductive age in Tanzania.

### Method

The study used data from the 2015–16 Tanzania Demographic and Health Survey and Malaria Indicators Survey (2015–16 TDHS-MIS). A total of 6924 women of active reproductive age from 15 to 49 were included in the analysis. Both univariate and multiple regression analyses were used to determine predictors of early antenatal booking.

### Results

Only 1586 (22.9%) of pregnant women had early antenatal booking. After adjusted for the confounders, factors which influenced early antenatal booking were age of a woman [20 to 34 years, Adjusted Odds Ratio (AOR) = 1.554 at 95% Confidence Interval (CI) = 1.213–1.993, and more than 34 years, AOR = 1.758 at 95% CI = 1.306–2.368]; wealth status [rich, AOR = 1.520 at 95% CI = 1.282–1.802]; education level [higher education, AOR = 2.355 at 95% CI = 1.36–4.079]; parity [Para 2 to 3, AOR = 0.85 at 95% CI = 0.727–0.994 and Para 5 +, AOR = 0.577 at 95%CI = 0.465–0.715]; zones [Unguja, AOR = 0.433 at 95% CI = 0.284–0.658 and Pemba, AOR = 0.392 at 95% CI = 0.236–0.649].

### Conclusion

Early antenatal booking in Tanzania is extremely low. Women who were more likely to initiate antenatal visits within the first 12 weeks are those from well-off families, those with

**Data Availability Statement:** Data are available from the Demographic and Health Survey program

(https://dhsprogram.com/data/) Tanzania 2015-16 data set.

**Funding:** The study was not funded.

**Competing interests:** No competing interest.

higher education, primiparity women, and those from Tanzania mainland urban. The innovative interventional study is highly recommended to come up with an effective strategy to improve timing for antenatal booking.

## Introduction

Maternal and neonatal morbidity and mortality remain to be a public health challenge worldwide. According to the World Health Organization (WHO) report of 2019, approximately 810 maternal death occurred every day in 2017 [1] of which 94% occurred in middle and low-income countries. Also, in the same year, 18 neonatal deaths occurred in every 1000 live birth worldwide. Tanzania is among the countries with the highest neonatal mortalities, in 2017, 21 neonatal deaths occurred in every 1000 live births [2]. Most of these deaths are due to preventable causes.

For every maternal or neonatal death, there are several morbidities which impair the quality of life of both mothers and children. Maternal morbidity refers to any physical or mental illnesses caused by pregnancy and/or childbirth [3]. Some of these illnesses are severe and can even lead to long term mental and physical disabilities [3]. Severe Maternal Morbidity (SMM) can be defined as potentially life-threatening conditions that may occur during pregnancy, childbirth, or after the termination of pregnancy [4]. The indicators for SMM are the presence of hemorrhagic disorders, hypertensive disorders, and other systemic disorders caused by pregnancy and/or childbirth [4]. The highest burden of severe maternal morbidity is in sub-Saharan Africa which is as high as 198 per 1000 live births [5].

Neonatal health depends on the health status and nutritional status of their mothers and the health care these mothers receive during pregnancy, childbirth, and immediately after birth [6]. Neonatal health problems are prematurity, low birth weight, infection, jaundice, and asphyxia [7]. All of these health problems are preventable if invested in the health of a woman before pregnancy, during pregnancy, and during childbirth. A previous study done in Kenya found a significant relationship between inadequate antenatal services utilization and neonatal morbidities and mortalities [8].

It is globally accepted that the use of antenatal services remains to be a cornerstone strategy towards the reduction of maternal and neonatal morbidities and mortalities. Although, life-threatening conditions cannot be predicted during ANC visits, the visit plays a key role in imparting pregnant women with essential knowledge on key danger signs, provision of nutrient supplements (iron and folic acids), treatment and prevention of diseases (deworming and intermittent treatment of malaria) and vaccination against tetanus. Empowering pregnant women with knowledge on obstetric danger signs is an important strategy towards timely obstetric care [9, 10]. Obstetric complications if timely attended can serve the lives of both mothers and children.

Tanzania like other developing countries adopted the Focused Antenatal Care (FANC) model which is four visits outlined by WHO as a strategy towards the reduction of maternal and neonatal deaths [11]. The first visit is between 8th and 12th gestation weeks where pregnancy is confirmed and expected date of delivery is calculated, classify a pregnant woman into either to attend the basic four or need specialized care, screen, treat and receive preventive measures and develop a birth plan. The second visit is between the 24th gestation week and 26th week, a pregnant woman is assessed for maternal and fetal well beings, to rule-out pregnancy-induced hypertension (PIH) and anemia, receive preventive measures, and review the

birth plan initiated in the first visit. The third visit is at 32 gestation weeks where a pregnant woman is assessed for maternal and fetal well beings, rules out PIH and anemia, assessed for multiple pregnancies, and receives preventive measures, and reviews the birth plan. The fourth visit is between the 36th week and 38th gestation week, a pregnant woman in this visit is assessed for maternal and fetal well beings, exclude PIH, anemia, multiple pregnancy and mal-presentation, receive preventive measures, and review the birth plan [11].

The majority of pregnant women (69%) in Africa make at least one ANC visit in the course of their pregnancy [11]. In Tanzania specifically almost all (98%) pregnant women make at least one antenatal visit [12]. A systematic review done in developing countries found out that maternal education, husband's education, marital status, availability, cost, household income, women's employment, media exposure and having a history of obstetric complications, cultural beliefs, parity, women's were factors which influence antenatal services utilization [13, 14].

Most pregnant women initiate antenatal visit late and make less than the required four visits. Early antenatal booking refers to the initiation of antenatal visits within the first three months of pregnancy. According to the FANC model, it is the first visit between 8th and 12th gestation weeks. Early initiation of the antenatal clinic allows early detection, managing, and prevents problems that occur during pregnancy time [14]. Previous studies have reported the significant relationship between late antenatal booking and pregnancy complications [15].

The FANC-ANC visits model has been in operation since 2002 [15] and the ANC sessions do encourage early initiation and adequate ANC visits but one wonders why this has been ignored this long. The study aimed at finding factors which influence early antenatal booking in Tanzania.

## Methods

### Study area and period

The study was conducted in the United Republic of Tanzania from August 22, 2015, through February 14, 2016. Tanzania is the largest country in East Africa which covers 940,000 square kilometers and 60,000 square kilometers is inland water. The country lies south of the equator and shares borders with eight countries: Kenya and Uganda to the North; Rwanda, Burundi, the Democratic Republic of Congo, and Zambia to the West; and Malawi and Mozambique to the South.

### Study design

It was a national-based cross-sectional study utilizing the 2015–16 Tanzania Demographic and Health Survey and Malaria Indicator Survey (TDHS-MIS) dataset.

The method section has been published earlier in the report of "Tanzania Demographic and Health Survey and Malaria Indicator Survey 2015–16" [12] The survey was coordinated and the report writing was coordinated by the National Bureau of Statistics (NBS) and the Office of Chief Government Statistician (OCGS), Zanzibar, in collaboration with the Ministry of Health, Community Development, Gender, Elderly, and Children on the Tanzania Mainland and the Ministry of Health, Zanzibar. The primary aim of the 2015–16 TDHS-MIS was to provide up-to-date estimates of basic demographic and health indicators. The data collected during the survey were fertility levels, marriage, sexual activity, fertility preferences, awareness and use of family planning methods, breastfeeding practices, nutrition, childhood and maternal mortality, maternal and child health, malaria, and other health-related issues.

## Sampling technique

Two stages of sampling were used to obtain a sample for urban and rural areas in Tanzania Mainland and Zanzibar. In the first stage, a total of 608 clusters were selected and in the second stage, a systematic selection of households was involved. In the second stage, a systematic selection of households was involved. A total of 22 households were then systematically selected from each cluster, yielding a representative probability sample of 13,376 households for the 2015–16 TDHS-MIS. To enhance representativeness Tanzania was divided into nine geographic zones. Grouping the regions into zones was done to reduce sampling error by increasing the number of people in the denominator. The zone was western (Tabora and Kigoma regions), Northern zone (Kilimanjaro, Tanga, and Arusha), Central zone (Dodoma, Singida and Manyara), Southern Highland zone (Iringa, Njombe, and Iringa), Southern zone (Lindi and Mtwara), South West Highland zone (Mbeya Rukwa and Katavi), Lake zone (Kagera, Mwanza, Geita, Mara, Simiyu, and Shinyanga), Eastern zone (Dar es Salaam, Pwani, and Morogoro) and Zanzibar (Kaskazini Unguja, Kusini Unguja, Mjini Magharibi, Kaskazini Pemba and Kusini Pemba).

## Data collection tool

The 2015–16 TDHS-MIS used household questionnaires and individual questionnaires. These questionnaires are based on the Measure DHS standard AIDS Indicator Survey and Malaria Indicator Survey questionnaires standards. They were adapted and modified to reflect the Tanzanian population. They were translated into Kiswahili, Tanzania's national language. The data presented in this study are from the individual questionnaire.

## Study population and data extraction

In the current study, the subset of the original TDHSMIS dataset was distracted using the criteria of women of reproductive age (15–49 years) who gave birth within five years preceding the survey. All other variables were dropped and a total of 6924 women were included in the study.

## Study variables

Through a literature review, the conceptual framework was developed to guide the conceptualization (Fig 1). The conceptual framework had primary independent variables (socio-demographic, obstetric characteristics of a woman, and women's perception towards early booking), the outcome variable was antenatal booking (Early or late). The outcome variable was a dummy variable coded as 1 if women had early antenatal booking and 0 if otherwise.

## Data analysis

Data were analyzed using IBM SPSS version 20. Data analysis started by describing all study variables using frequencies and percentages. The associations between the independent variable were established using chi-squire analysis. Variables that showed a significant relationship with the dependent variable were entered in the regression model to assess their effect on the outcome variable. A binary logistic regression analysis was performed (univariate and multivariable) to determine predictors of early antenatal booking. All analyses were based at a 5% level of significance.

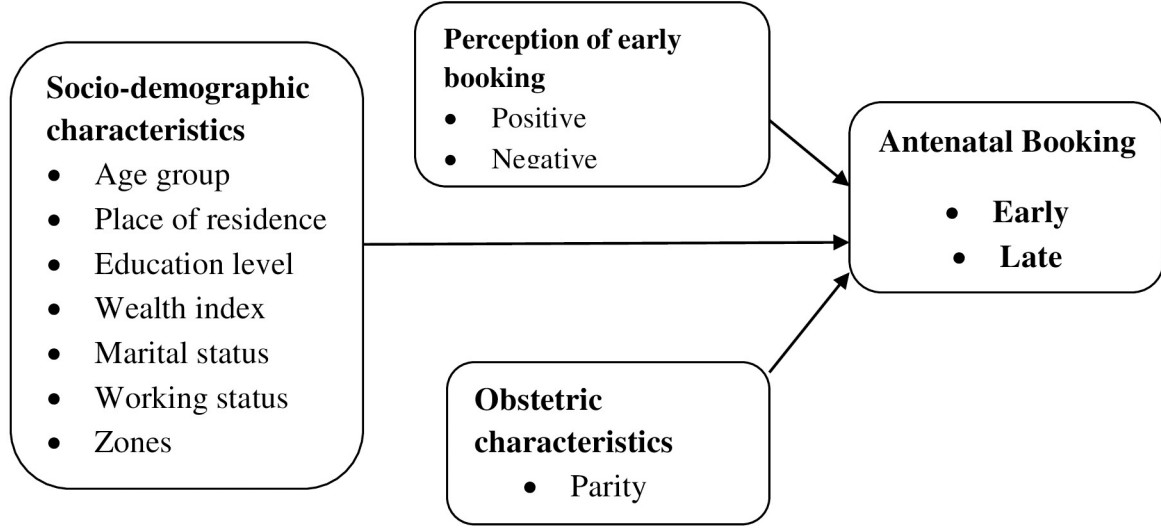

**Fig 1. Conceptual framework on factors influencing early antenatal booking.**

### Ethics approval and consent to participate

This study used secondary data without involving any human subjects which therefore did not require formal ethical approval. Data were fully anonymized before the researcher accessed them. The request to use the data was sought from DHS MEASURES. Permission was given subject to using the data for this particular research topic only and publishing the findings in a peer-reviewed journal.

## Results

### Socio-demographic characteristics

The study included 6924 women of reproductive age who had given birth within five years preceding the survey. The majority of study respondents 5113(73.8%) resided in the rural setting of Tanzania, aged 20 to 34 years 4557(65.8%), had primary education 4209(60.8) and were married 5650(86.1%) (Table 1).

### The relationship between socio-demographic characteristics and early antenatal booking

Early antenatal booking had a significant relationship with the place of residence (p<0.001), age group of a woman (p<0.001), education level of a woman (p<0.001), parity (p<0.001), wealth index (p<0.001) Table 2.

### Factors which influence early antenatal booking

After adjusted for the confounders, factors which influenced early antenatal booking were age of a woman [20 to 34 years, Adjusted Odds Ratio (AOR) = 1.554 at 95% Confidence Interval (CI) = 1.213–1.993, p = 0.001 and more than 34 years, AOR = 1.758 at 95% CI = 1.306–2.368, p<0.001]; wealth status [rich, AOR = 1.520 at 95% CI = 1.282–1.802, p<0.001]; education level [higher education, AOR = 2.355 at 95% CI = 1.36–4.079, p = 0.002]; parity [para 2 to 3, AOR = 0.85 at 95% CI = 0.727–0.994, p = 0.041, para 5+, AOR = 0.577 at 95%CI = 0.465–

**Table 1. Socio-demographic characteristics (n = 6924).**

| Variables | Frequency | Percent (%) |
|---|---|---|
| **Place of residence** | | |
| Urban | 1811 | 26.2 |
| Rural | 5113 | 73.8 |
| **Age group** | | |
| Less than 20 years | 541 | 7.8 |
| 20 to 34 years | 4557 | 65.8 |
| More than 34 years | 1826 | 26.4 |
| **Educational level** | | |
| No education | 1329 | 19.2 |
| Primary education | 4209 | 60.8 |
| Secondary | 1326 | 19.2 |
| Higher | 60 | 0.9 |
| **Parity** | | |
| Para one | 1595 | 23 |
| Para 2–4 | 3154 | 45.6 |
| Para 5+ | 2175 | 31.4 |
| **Wealth index** | | |
| Poor | 2734 | 39.5 |
| Middle | 1363 | 19.7 |
| Rich | 2827 | 40.8 |
| **Marital Status** | | |
| Never in union | 441 | 6.4 |
| Married | 5650 | 86.1 |
| Widow | 119 | 1.7 |
| Separated | 714 | 10.3 |
| **Respondent currently working** | | |
| Not working | 1498 | 21.6 |
| Working | 5426 | 78.4 |
| **Mainland/Zanzibar** | | |
| Mainland urban | 1618 | 23.4 |
| Mainland rural | 4357 | 62.9 |
| Unguja (Zanzibar Island) | 594 | 8.6 |
| Pemba (Pemba Island) | 355 | 5.1 |
| **Women perception towards early ANC booking** | | |
| Positive | 6873 | 99.3 |
| Negative | 51 | 0.7 |

0.715, p<0.001]; zones [Unguja, AOR = 0.433 at 95% CI = 0.284–0.658, p<0.001 and Pemba, AOR = 0.392 at 95% CI = 0.236–0.649, p<0.001] (Table 3).

## Discussion

Although early antenatal booking provides early detection, management, and prevention of problems that may occur during pregnancy time, the prevalence of early antenatal initiation in Tanzania is extremely low (22.9%). The WHO has recommended folic acid supplementation as early as possible in pregnancy to prevent neural tube defects [13]. The first 12 weeks of pregnancy is a vital period for a neonate when the organogenesis is taking place. It is a critical time when folic acid supplementation is needed. The folic acid supplementation is highly needed in

**Table 2. The relationship between socio-demographic characteristics and antenatal booking (n = 6924).**

| Variables | ANC Booking | | | |
|---|---|---|---|---|
| | Early | Late | X2 | p-value |
| **Place of residence** | | | | |
| Urban | 527(29.1) | 1284(70.9) | | |
| Rural | 1059(20.7) | 4054(79.3) | 53.283 | <0.001 |
| **Age group** | | | | |
| Less than 20 years | 102(18.9) | 439(81.1) | | |
| 20 to 34 years | 1109(24.3) | 3448(75.7) | | |
| More than 34 years | 375(20.5) | 1451(79.5) | 16.113 | <0.001 |
| **Educational level** | | | | |
| No education | 251(18.9) | 1078(81.1) | | |
| Primary education | 979(23.3) | 3230(76.7) | | |
| Secondary | 327(24.7) | 999(75.3) | | |
| Higher | 29(48.3) | 31(51.7) | 36.737 | <0.001 |
| **Parity** | | | | |
| Para one | 423(26.5) | 1172(73.5) | | |
| Para 2–4 | 780(24.7) | 2374(75.3) | | |
| Para 5+ | 383(17.6) | 1792(82.4) | 52.3 | <0.001 |
| **Wealth index** | | | | |
| Poor | 520(19) | 2214(81) | | |
| Middle | 290(21.3)_ | 1073(78.7) | | |
| Rich | 776(27.4) | 2051(72.6) | 58.481 | <0.001 |
| **Marital Status** | | | | |
| Never in union | 108(75.5) | 333(75.5) | | |
| Married | 1260(22.3) | 4390(77.7) | | |
| Widow | 32(26.9) | 87(73.1) | | |
| Separated | 186(26.1) | 528(73.9) | 6.866 | 0.076 |
| **Respondent currently working** | | | | |
| Not working | 341(22.8) | 1157(77.2) | | |
| Working | 1245(22.9) | 4181(77.1) | 0.022 | 0.882 |
| **Mainland/Zanzibar** | | | | |
| Mainland urban | 496(30.7) | 1122(69.3) | | |
| Mainland rural | 944(21.7) | 3413(78.3) | | |
| Unguja (Zanzibar Island) | 99(16.7) | 495(83.3) | | |
| Pemba (Pemba Island) | 47(13.2) | 308(86.8) | 90.691 | <0.001 |
| **Women perception towards early ANC booking** | | | | |
| Positive | 1581(23) | 5292(77) | | |
| Negative | 5(9.8) | 46(90.2) | 4.994 | 0.025 |

low resources countries like Tanzania where women become pregnant while anemic and with micronutrient deficient.

A bit higher prevalence was reported in similar studies done elsewhere in Africa [14–16]. The observed difference could be due to the differences in the sample size involved and the criterion used to categorize early booking. While this study used population-based data with large sample size, other studies used small samples. Also, the criteria used to categorize early booking differ; early booking in this study was antenatal visit within twelve weeks while other studies went as far as 16 weeks and 20weeks. The controversy on early antenatal booking is now cleared with the release of a new model for antenatal attendance by the World Health

**Table 3. Predictors of early antenatal booking among women of reproductive age in Tanzania (n = 6924).**

| Variable | OR | 95%CI | | p-value | AOR | 95%CI | | p-value |
|---|---|---|---|---|---|---|---|---|
| | | Lower | Upper | | | Lower | Upper | |
| **Age groups** | | | | | | | | |
| Less than 20 years | 1 | | | | 1 | | | |
| 20 to 34 years | 1.384 | 1.104 | 1.735 | 0.005 | 1.554 | 1.213 | 1.993 | 0.001 |
| More than 34 years | 1.112 | 0.872 | 1.419 | 0.392 | 1.758 | 1.306 | 2.368 | <0.001 |
| **Place of residence** | | | | | | | | |
| Urban | 1 | | | | 1 | | | |
| Rural | 0.636 | 0.563 | 0.719 | <0.001 | 1.122 | 0.721 | 1.745 | 0.611 |
| **Wealth index** | | | | | | | | |
| Poor | 1 | | | | 1 | | | |
| Middle | 1.151 | 0.98 | 1.352 | 0.088 | 1.157 | 0.981 | 1.364 | 0.082 |
| Rich | 1.611 | 1.42 | 1.828 | <0.001 | 1.520 | 1.282 | 1.802 | <0.001 |
| **Educational level** | | | | | | | | |
| No education | 1 | | | | 1 | | | |
| Primary education | 1.302 | 1.115 | 1.52 | 0.001 | 1.037 | 0.881 | 1.22 | 0.664 |
| Secondary | 1.406 | 1.168 | 1.693 | <0.001 | 1.018 | 0.818 | 1.266 | 0.873 |
| Higher | 4.018 | 2.378 | 6.789 | <0.001 | 2.355 | 1.36 | 4.079 | 0.002 |
| **Parity** | | | | | | | | |
| Para one | 1 | | | | | | | |
| Para 2–4 | 0.91 | 0.793 | 1.045 | 0.18 | 0.85 | 0.727 | 0.994 | 0.041 |
| Para 5+ | 0.592 | 0.506 | 0.693 | <0.001 | 0.577 | 0.465 | 0.715 | <0.001 |
| **Mainland/Zanzibar** | | | | | | | | |
| Mainland urban | 1 | | | | 1 | | | |
| Mainland rural | 0.626 | 0.551 | 0.711 | <0.001 | 0.807 | 0.507 | 1.284 | 0.365 |
| Unguja (Zanzibar Island) | 0.452 | 0.356 | 0.575 | <0.001 | 0.433 | 0.284 | 0.658 | <0.001 |
| Pemba (Pemba Island) | 0.345 | 0.25 | 0.478 | <0.001 | 0.392 | 0.236 | 0.649 | <0.001 |
| **Women perception towards early ANC booking** | | | | | | | | |
| Positive | 1 | | | | 1 | | | |
| Negative | 0.364 | 0.144 | 0.917 | 0.032 | 0.444 | 0.175 | 1.124 | 0.086 |

Organization. The new model stipulates clearly that the first antenatal booking should be made within the first 12 weeks of pregnancy [9].

This study found that one of the predictors of early antenatal booking was the age of the pregnant woman. The likelihood of initiating antenatal visits early increased with an increase in age. Pregnant women who were aged 20 to 34 years were 1.5 times more likely to initiate antenatal visits early compared to pregnant women who were aged less than 20 years. Also, those aged 35 years and above were 1.7 times more likely to have an early antenatal booking. This is a surprising finding because it is expected that younger women who are inexperienced and are in their first pregnancies could book earlier than older women. A previous study done in Tanzania has reported that being single increases the likelihood of late booking [17]. The majority of pregnant women younger than 20 years are single. This could be the reason for low odds in early antenatal booking among pregnant women of less than 20 years of age. Also, the majority of ANC clinics in Tanzania are not youth-friendly. It is difficult for a youth to attend a clinic where she will meet her aunts and mothers. For a young woman to be pregnant without marriage is against both their religion believes and their cultural believes. The study recommends to the government of Tanzania through its ministry of health to have youth-friendly reproductive health centers. The different finding was reported by a previous study

done in Ethiopia which showed older women were 3 times more likely to initiate antenatal visits late compared to younger women [18]. The difference could be due to different study populations.

The study also found that level of education of pregnant women significantly influenced timing for antenatal initiation. Pregnant women with higher education were 2.4 more likely to initiate antenatal early compared to those with no formal education. Formal education could have exposed them to health-related information and sensitize them on maternal services utilization. It is recommended that the health information provided in antenatal sessions focus more on illiterate women to sensitize them to use maternal services. Similar studies have reported similar findings [14, 19].

The parity of a woman also significantly influenced timing for antenatal initiation. Pregnant women of higher parity were less likely to initiate antenatal sessions early. The likelihood decreased as the parity increased. Pregnant women of 2 to three parity were 15% less likely to initiate antenatal visits early when compared to primipara and those with 5 and more children were 42.3% less likely to initiate antenatal visits within the first 12 weeks of pregnancy. Similar findings were reported by similar previous studies [19, 20]. Pregnant women of high parity are most likely to have low-risk perception towards pregnancy and childbirth [21]. The low-risk perception could be influenced by uneventful previous childbirth. This is worrisome because each pregnancy can develop serious complications. The study recommends for needful effort on sensitizing multiparity women on risks associated with pregnancy and childbirth. Previous studies have reported that pregnant women who had complications in their previous pregnancy are more likely to initiate antenatal clinic early if compared to those without complications.

The study also found a significant association between antenatal initiation and wealth status. Pregnant women from rich families were 1.5 times more likely to initiate antenatal clinic early if compared to pregnant women from poor families. Antenatal services in Tanzania are offered for free but there are hidden costs that pregnant women and her family have to incur to receive the service. This is evidenced by a previous study that found a significant association between early initiation of antenatal clinics and distance to a nearby health facility [15]. The longer the distance the higher the transport cost.

Pregnant women from the Zanzibar islands were significantly less likely to initiate antenatal booking early if compared to pregnant women in mainland urban. Those from Unguja Island were 56.7% less likely to initiate antenatal clinics early if compared to pregnant women from mainland urban. Pregnant women from Pemba Island were 60.8% less likely to initiate antenatal clinics early if compared with pregnant women from mainland urban. The significance of low odds to initiate antenatal clinic timely in Zanzibar and Pemba Islands could be due to religious factors. A systematic review done in the sub-Saharan region has revealed that being Christian increases the likelihood to initiate ANC clinic timely compared to other religions [22]. Majority of the population in Zanzibar and Pemba Island are Muslims compared to Mainland Urban where the majority are Christians. The study recommends a deliberate effort to increase the early initiation of antenatal clinics in this population.

Also, pregnant women from mainland rural were 19.3% less likely to initiate antenatal clinics early compared to mainland urban but the difference was not statistically significant. Despite no significant association, previous studies have reported urban dwellers to initiate ANC early compared to rural dwellers [22]. It is evidenced that pregnant women who know when to initiate ANC are more likely to do so if compared to those who do not know [22]. The study urges that the government of Tanzania through its ministry of Health strengthen community sensitization on early antenatal booking with more focus on women from rural communities.

This study was not without limitations, it used quantitative methods which finds only the association between variables but do little to explain them. A corresponding qualitative study is recommended to explain the association. Also, the study was a population-based survey which cannot explain the causal effect relationship; it can only explain the existence of a relationship between variables. An experimental design study design is recommended for the causal-effect relationship of variables.

## Conclusion

Early antenatal booking in Tanzania is extremely low. Women who are less likely to initiate antenatal visits within the first 12 weeks are those from poor families, of low education, high parity, and from Zanzibar Islands. The innovative interventional study is highly recommended to come up with an effective strategy to improve timing for antenatal booking.

## Acknowledgments

The author is grateful to MEASURE DHS for providing them with the data set.

## Author Contributions

**Conceptualization:** Fabiola V. Moshi.

**Formal analysis:** Fabiola V. Moshi.

**Validation:** Fabiola V. Moshi.

**Writing – original draft:** Fabiola V. Moshi.

**Writing – review & editing:** Fabiola V. Moshi.

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
