## [Decision Letter · Decision Letter 0]

17 Nov 2020

PONE-D-20-15843

Prevalence and Factors which Influence Early Antenatal Booking Among Women of Reproductive Age in Tanzania: An Analysis of Data from the 2015-16 Tanzania HIV and Malaria Indicators Survey

PLOS ONE

Dear Dr. Moshi,

Thank you for submitting your manuscript to PLOS ONE. After careful consideration, we feel that it has merit but does not fully meet PLOS ONE’s publication criteria as it currently stands. Therefore, we invite you to submit a revised version of the manuscript that addresses the points raised during the review process.

Two experts have reviewed your manuscript and have several concerns that need to be addressed. Please see their review comments below.

We look forward to receiving your revised manuscript.

Kind regards,

Susan Hepp

Academic Editor

PLOS ONE

Journal Requirements:

2. In your ethics statement in the Methods section and in the online submission form, please clarify whether all data were fully anonymized before you accessed them.

In addition, PLOS ONE requires statistical methods to be described in enough detail to allow suitably skilled investigators to fully replicate and evaluate a study. After internal review, we were concerned that the experimental and statistical methods may not detailed enough to meet these criteria.

Please revise your manuscript to include a more detailed description of your statistical methods.

Our submission requirements can be found here:  https://journals.plos.org/plosone/s/submission-guidelines#loc-statistical-reporting

Reviewers' comments:

Reviewer's Responses to Questions

**Comments to the Author**

1. Is the manuscript technically sound, and do the data support the conclusions?

Reviewer #1: Partly

Reviewer #2: Yes

2. Has the statistical analysis been performed appropriately and rigorously? 

Reviewer #1: No

Reviewer #2: Yes

3. Have the authors made all data underlying the findings in their manuscript fully available?

Reviewer #1: No

Reviewer #2: Yes

4. Is the manuscript presented in an intelligible fashion and written in standard English?

Reviewer #1: No

Reviewer #2: No

5. Review Comments to the Author

Reviewer #1: The article contains some interesting information regarding factors related to antenatal care visits in underserved settings, but the reviewer would like to make some comments before it should be considered for publication.

Major comments

1. In the Introduction, the author needs to fully and thoroughly review previous research regarding antenatal care (ANC) visits. There are several review articles related to positive and negative factors of ANC participation. In both high and low-middle income countries, low maternal education, non-marital status, and high parity are associated with inadequate use of ANC (Simkhada, 2008; Feijen-de Jong, 2012). Then, the author needs to describe why the study on Tanzania is required at this time.

2. The reviewer does not clearly make the connection between the last sentence “little is known …”, in the last paragraph and the second to last paragraph in the Introduction. Also, there is no statement of the objectives of the study.

3. The author stated that the WHO recommended antenatal model including numbers of ANC visit in 2016. However, the author uses data collected in 2015-2016. The reviewer assumes that ANC provision may have followed the previous WHO recommendation at the time of data collection in Tanzania. It may be reasonable to evaluate data in 2015-2016 according to the previous WHO recommendation and the national guideline of ANC at the time of data collection. In addition, more detail of ANC related conditions in Tanzania, which are distribution of healthcare personnel and healthcare facilities, and quantity and quality of ANC provision, will be informative for readers of this manuscript, if the author adds them in the Introduction section.

4. If the author provided an association between religion and ANC visits, it would be informative, because early ANC visits were less likely to be observed in Zanzibar than mainland, where there are more Muslims.

5. If the author considered confounders and/or mediators among independent variables, it would be informative to the interpretation of the findings.

6. The author mentioned that teenage pregnant women are less likely to initiate ANC visit than the older pregnant women. Is there a possibility of unplanned teenage pregnancy without marriage? How about conditions for and provision of youth friendly services in Tanzania? How about cultural and/or social norms regarding teenage pregnancy? The reviewer considers marital status, cultural and/or social norms, and access to and quality of youth friendly services provision are important factors when evaluating maternity services including ANC and consequences/outcomes among teenage pregnancy.

7. In the last paragraph of the Discussion section, the author should discuss in-depth the Zanzibar conditions from the perspective of religious, cultural and/or social norms, distribution of healthcare personnel, and quantity and quality of ANC service provision.

8. It would be better to describe the limitations of this study in the Discussion section.

Minor comments

1. Abstract: In the Results section, AOR and CI should be spelled out when used for the first time.

2. Results section: In the “Factors which influence early antenatal booking”, AOR and CI should be spelled out when used at the first time.

3. Discussion section: The last sentence of the third to the last paragraph needs referencing. ‘Previous studies have reported that pregnant women ….. if compared with those without complication.’

4. Overall, the reviewer recommends basic editing of the article, such as adjusting spacing between word and word, word and brackets, and after comma, because there are unnecessary spaces and/or no spaces.

Reviewer #2: REVIEWER COMMENTS

Abstract: Minor grammatical errors to be corrected

Introduction: Requires cleaning up. The literature review in the introduction section needs to be summarized especially the area on WHO 2016 report. It is more of copy and paste.

Methods

Study population: Conflicting figures from the previous statement

Sampling technique: The sampling technique is faulty. What you did is a multi-stage sampling and not two stages. Tanzania was divided into 8 zones. This should have formed your clusters or strata from where you will select the households. How did you arrive at 608 clusters?

How you reached your study participants after selecting the households was not stated

Data collection tool: Was the translated questionnaire to Kiswahili Tanzania’s national language validated? If not it is a flaw in this study

How you administered the questionnaire to the study participants was not expla9ined

Data analysis: Recast the first sentence

Results: What happened to 117 of the study participants?

Discussions: Requires cleaning up for grammatical errors

Conclusion: What you did is not an interventional study so delete

Declarations: The questionnaire was used as interview guide to interview your study participants.

References : Revisit to be consistent,

6. PLOS authors have the option to publish the peer review history of their article (what does this mean?). If published, this will include your full peer review and any attached files.

Reviewer #1: No

Reviewer #2: No

---

## [Author Response · Author response to Decision Letter 0]

25 Jan 2021

RESPONSE TO REVIEWERS'. COMMENTS

Editor’s comments

Comment 1

Please ensure that your manuscript meets PLOS ONE's style requirements, including those for file naming. The PLOS ONE style templates can be found at https://journals.plos.org/plosone/s/file?id=wjVg/PLOSOne_formatting_sample_main_body.pdf and https://journals.plos.org/plosone/s/file?id=ba62/PLOSOne_formatting_sample_title_authors_affiliations.pdf

Response 1

The manuscript meets PLOS ONE style

Comment 2

In your ethics statement in the Methods section and in the online submission form, please clarify whether all data were fully anonymized before you accessed them.

In addition, PLOS ONE requires statistical methods to be described in enough detail to allow suitably skilled investigators to fully replicate and evaluate a study. After internal review, we were concerned that the experimental and statistical methods may not detailed enough to meet these criteria.

Please revise your manuscript to include a more detailed description of your statistical methods.

Response 2

Data were extracted from the original file where information which were not a focus of the study were dropped. Also, the dataset was cleaned by removing incomplete cases; line 191 to 193

Method section is revised and description of data analysis is included line 205 to 208

Comment 3

Your ethics statement should only appear in the Methods section of your manuscript. If your ethics statement is written in any section besides the Methods, please move it to the Methods section and delete it from any other section. Please ensure that your ethics statement is included in your manuscript, as the ethics statement entered into the online submission form will not be published alongside your manuscript.

Response 3

Author agree with the comment and ethical statement is moved to the method section

Reviewers' comments:

Reviewer 1

Comment 1

In the Introduction, the author needs to fully and thoroughly review previous research regarding antenatal care (ANC) visits. There are several review articles related to positive and negative factors of ANC participation. In both high and low-middle income countries, low maternal education, non-marital status, and high parity are associated with inadequate use of ANC (Simkhada, 2008; Feijen-de Jong, 2012). Then, the author needs to describe why the study on Tanzania is required at this time.

Response 1

The introduction is reviewed and the suggested literature were visited and used in the introduction Line 76 to line 135

Comment 2

The reviewer does not clearly make the connection between the last sentence “little is known …”, in the last paragraph and the second to last paragraph in the Introduction. Also, there is no statement of the objectives of the study.

Response 2

The two paragraphs are re-written and the line of argument is connected

Comment 3

The author stated that the WHO recommended antenatal model including numbers of ANC visit in 2016. However, the author uses data collected in 2015-2016. The reviewer assumes that ANC provision may have followed the previous WHO recommendation at the time of data collection in Tanzania. It may be reasonable to evaluate data in 2015-2016 according to the previous WHO recommendation and the national guideline of ANC at the time of data collection. In addition, more detail of ANC related conditions in Tanzania, which are distribution of healthcare personnel and healthcare facilities, and quantity and quality of ANC provision, will be informative for readers of this manuscript, if the author adds them in the Introduction section.

Response 3

Author agrees with the comment and the WHO recommendation presented in the introduction is the one used during the time of survey line 84 to line 97

Comment 4

If the author provided an association between religion and ANC visits, it would be informative, because early ANC visits were less likely to be observed in Zanzibar than mainland, where there are more Muslims.

Response 4

Author agree with the observation and explanation on to why there are low proportion of early booking among pregnant women in Zanzibar Islands is provided, line 337 to line 344

Comment 5

If the author considered confounders and/or mediators among independent variables, it would be informative to the interpretation of the findings.

Response 5

Comment 6

The author mentioned that teenage pregnant women are less likely to initiate ANC visit than the older pregnant women. Is there a possibility of unplanned teenage pregnancy without marriage? How about conditions for and provision of youth friendly services in Tanzania? How about cultural and/or social norms regarding teenage pregnancy? The reviewer considers marital status, cultural and/or social norms, and access to and quality of youth friendly services provision are important factors when evaluating maternity services including ANC and consequences/outcomes among teenage pregnancy.

Response 6

The reasons for teenage pregnant women to initiate ANC clinics late is added in line 302 to line 307. Marital status was among the variables which were considered to influence early antenatal booking. The variable did not show significant relationship with ANC booking in the cross tabulation and then it was not taken to regression analysis. 

Comment 7

In the last paragraph of the Discussion section, the author should discuss in-depth the Zanzibar conditions from the perspective of religious, cultural and/or social norms, distribution of healthcare personnel, and quantity and quality of ANC service provision.

Response 7

The message is included in the discussion

Comment 8

It would be better to describe the limitations of this study in the Discussion section.

Response 8

A paragraph on limitation of the study is added line 363-368

Comment 9

Abstract: In the Results section, AOR and CI should be spelled out when used for the first time.

Response 9

The two abbreviations are now spelled out in the first time used in the abstract line 36.

Comment 10

Results section: In the “Factors which influence early antenatal booking”, AOR and CI should be spelled out when used at the first time.

Response 10

The two abbreviations are now spelled out in the first time used in the results section line 248.

Comment 11

Discussion section: The last sentence of the third to the last paragraph needs referencing. ‘Previous studies have reported that pregnant women ….. if compared with those without complication.’

Response 11

The sentence is re-phrased line 334

Comment 12

Overall, the reviewer recommends basic editing of the article, such as adjusting spacing between word and word, word and brackets, and after comma, because there are unnecessary spaces and/or no spaces.

Response 12

Authors agree with the recommendation and the editing was done

Reviewer 2

Comment 1

Abstract: Minor grammatical errors to be corrected

Response 1

The grammatical errors are corrected

Comment 2

Introduction: Requires cleaning up. The literature review in the introduction section needs to be summarized especially the area on WHO 2016 report. It is more of copy and paste.

Response 2

The introduction is re-written and paraphrasing of reference documents used is done

Comment 3

Methods 

i) Study population: Conflicting figures from the previous statement

Response

The figure is corrected

ii) Sampling technique: The sampling technique is faulty. What you did is a multi-stage sampling and not two stages. Tanzania was divided into 8 zones. This should have formed your clusters or strata from where you will select the households. How did you arrive at 608 clusters?

How you reached your study participants after selecting the households was not stated

Response 

The study used secondary data. The method section has been published in the report of Tanzania Demographic and Health Survey and Malaria Indicator Survey 2015-2016

iii) Data collection tool: Was the translated questionnaire to Kiswahili Tanzania’s national language validated? If not it is a flaw in this study

How you administered the questionnaire to the study participants was not expla9ined

Response 

The study used secondary data. The method section has been published in the report of Tanzania Demographic and Health Survey and Malaria Indicator Survey 2015-2016

iv) Data analysis: Recast the first sentence

Response

The first sentence is re-casted

Comment 4

Results: What happened to 117 of the study participants?

Response 4

Study participants with missing information on timing for ANC visits were removed 

Comment 5

Discussions: Requires cleaning up for grammatical errors

Response 5

The grammatic errors corrected

Comment 6

Conclusion: What you did is not an interventional study so delete

Response 6

It is true what I did is not an intervention study but, in the conclusion, author is recommending for an intervention study which is more conclusive than a cross-sectional survey.

Comment 7

Declarations: The questionnaire was used as interview guide to interview your study participants.

Response 7

Yes, questionnaire was used to gather data but the author used the already developed data set

Comment 8

References: Revisit to be consistent,

Response 8

The list of references is revisited

---

## [Decision Letter · Decision Letter 1]

17 Mar 2021

Prevalence and Factors which Influence Early Antenatal Booking Among Women of Reproductive Age in Tanzania: An Analysis of Data from the 2015-16 Tanzania HIV and Malaria Indicators Survey

PONE-D-20-15843R1

Dear Dr. Fabiola V. Moshi,

We’re pleased to inform you that your manuscript has been judged scientifically suitable for publication and will be formally accepted for publication once it meets all outstanding technical requirements.

Kind regards,

Claudia Marotta

Academic Editor

PLOS ONE

Additional Editor Comments (optional):

dear authors congratulations

Reviewers' comments:

Reviewer's Responses to Questions

**Comments to the Author**

1. If the authors have adequately addressed your comments raised in a previous round of review and you feel that this manuscript is now acceptable for publication, you may indicate that here to bypass the “Comments to the Author” section, enter your conflict of interest statement in the “Confidential to Editor” section, and submit your "Accept" recommendation.

Reviewer #1: All comments have been addressed

2. Is the manuscript technically sound, and do the data support the conclusions?

Reviewer #1: Yes

3. Has the statistical analysis been performed appropriately and rigorously? 

Reviewer #1: Yes

4. Have the authors made all data underlying the findings in their manuscript fully available?

Reviewer #1: Yes

5. Is the manuscript presented in an intelligible fashion and written in standard English?

Reviewer #1: Yes

6. Review Comments to the Author

Reviewer #1: There are still inappropriate editing of the article, such as adjusting spacing between word and word, word and brackets, and after comma.

7. PLOS authors have the option to publish the peer review history of their article (what does this mean?). If published, this will include your full peer review and any attached files.

Reviewer #1: No

---

## [Editor Report · Acceptance letter]

23 Mar 2021

PONE-D-20-15843R1 

Prevalence and Factors which Influence Early Antenatal Booking Among Women of Reproductive Age in Tanzania: An Analysis of Data from the 2015-16 Tanzania HIV and Malaria Indicators Survey 

Dear Dr. Moshi:

I'm pleased to inform you that your manuscript has been deemed suitable for publication in PLOS ONE. Congratulations! Your manuscript is now with our production department. 

Kind regards, 

on behalf of

Dr. Claudia Marotta 

Academic Editor

PLOS ONE